# *Schizosaccharomyces pombe* Rtf2 is important for replication fork barrier activity of *RTS1* via splicing of *Rtf1*

Alice M Budden[1], Murat Eravci[2†], Adam T Watson[1], Eduard Campillo-Funollet[1‡], Antony W Oliver[1], Karel Naiman[1*§], Antony M Carr[1*]

[1]Genome Damage and Stability Centre, University of Sussex, Brighton, United Kingdom; [2]Department of Biochemistry and Biomedicine, University of Sussex, Brighton, United Kingdom

**\*For correspondence:**
karel.naiman@inserm.fr (KN);
A.M.Carr@sussex.ac.uk (AMC)

**Present address:** †Medicines and Healthcare Products Regulatory Agency (MHRA), Hertfordshire, United Kingdom; ‡Department of Mathematics and Statistics, Lancaster University, Lancaster, United Kingdom; §Cancer Research Center of Marseille (CRCM), INSERM U1068, CNRS, Aix Marseille Université, Institut Paoli-Calmettes, Marseille, France

**Competing interest:** The authors declare that no competing interests exist.

**Abstract** Arrested replication forks, when restarted by homologous recombination, result in error-prone DNA syntheses and non-allelic homologous recombination. Fission yeast *RTS1* is a model fork barrier used to probe mechanisms of recombination-dependent restart. *RTS1* barrier activity is entirely dependent on the DNA binding protein Rtf1 and partially dependent on a second protein, Rtf2. Human RTF2 was recently implicated in fork restart, leading us to examine fission yeast Rtf2's role in more detail. In agreement with previous studies, we observe reduced barrier activity upon *rtf2* deletion. However, we identified Rtf2 to be physically associated with mRNA processing and splicing factors and *rtf2* deletion to cause increased intron retention. One of the most affected introns resided in the *rtf1* transcript. Using an intronless *rtf1*, we observed no reduction in RFB activity in the absence of Rtf2. Thus, Rtf2 is essential for correct *rtf1* splicing to allow optimal *RTS1* barrier activity.

## Editor's evaluation

This valuable work reports a new regulation of the RNA splicing of a fission yeast gene necessary for the DNA replication fork barrier. Using molecular genetic approaches combined with replication fork profiling analysis, the authors provided significant evidence that the Rtf2 is a splicing regulator specific to the *rtf1* gene essential for the fork barrier. The study is of interest to researchers in the fields of DNA replication and homologous recombination as well as RNA splicing.

## Introduction

The completion of DNA synthesis is crucial for maintaining genome stability and survival but a variety of obstacles to DNA replication have the ability to stall replication forks (*Magdalou et al., 2014*). Stalled forks are usually stabilised by the Intra-S phase checkpoint such that they can ordinarily resume DNA synthesis once the obstacle has been removed or resolved (*Lambert and Carr, 2013*). However, if replication cannot be resumed and/or the replication fork becomes non-functional, this is known as replication fork collapse. In the majority of cases collapsed replication forks are rescued by a convergent fork, allowing the completion of DNA synthesis. Nevertheless, in regions of the genome with a paucity of origins, or when two convergent forks collapse without an intervening origin, collapsed replication forks must be actively restarted in order to complete replication.

Homologous recombination underpins several mechanisms that have evolved to restart collapsed replication forks. Recombination-dependent replication (RDR) mechanisms include the restart of replication forks following fork reversal plus DNA end processing (*Ait Saada et al., 2018*) and

break-induced replication (BIR; *Malkova and Ira, 2013*), where replication is initiated from one end of a DNA double strand break (DSB). Yeast model systems have been instrumental in identifying and characterising RDR mechanisms: BIR has been extensively characterised in *Saccharomyces cerevisiae* and has been shown to occur in G2 phase and to involve initial DSB processing followed by Rad51-dependent strand invasion that results in conservative DNA synthesis via a migrating D-loop. In contrast, DSB-independent RDR (*Mizuno et al., 2009*) has been characterised mainly in *Schizosaccharomyces pombe* and has been shown to involve fork reversal, processing of the resulting DNA double strand end and Rad51-dependent strand invasion that results in semi-conservative replication (*Teixeira-Silva et al., 2017*; *Miyabe et al., 2015*).

A key tool used in *S. pombe* to investigate the mechanisms involved in replication fork arrest, collapse and restart is a site-specific replication fork barrier (RFB) that was initially identified close to the mating type locus (*Dalgaard and Klar, 2000*). This RFB, known as *RTS1* (Replication Termination Sequence 1), acts to ensure that replication across the mating type locus is unidirectional. It achieves this by acting as a polar barrier – i.e. it only arrests replication forks travelling from a single 'restrictive' direction (*Dalgaard and Klar, 2001*). Forks travelling in the opposite 'permissive' direction are unaffected by the barrier. The *RTS1* barrier was first defined as an 859 bp *Eco*RI fragment that was further refined into two regions, A and B, by deletion analysis. Region B contains four repeat sequences that bind to a Myb-domain protein, known as Rtf1 (Replication Termination Factor 1). Fork arrest at *RTS1* is entirely dependent on Rft1 binding (*Eydmann et al., 2008*) and, in the absence of Rtf1, replication of the *RTS1* sequence is entirely normal. In contrast, region A is defined as being required for enhancing barrier activity. Loss of region A has been reported to reduce barrier activity by approximately three quarters and this function was described as being dependent on a second protein, Rtf2 (*Codlin and Dalgaard, 2003*).

Interestingly, while Rtf1 appears to be *S. pombe* specific and is not evolutionarily conserved beyond the Myb-like DNA binding domain (*Eydmann et al., 2008*), Rtf2 is conserved in many eukaryotes including humans (*Codlin and Dalgaard, 2003*). Rtf2 belongs to a family of proteins that are characterised by a C2HC2 Ring Finger motif predicted to fold up into a RING-finger structure with the ability to bind one $Zn^{2+}$ ion (*Inagawa et al., 2009*) and is likely to mediate protein-protein interactions. Studies in human cells have shown that *Hs*RTF2 acts to reduce the levels of replication fork restart and thus its removal from arrested replication forks via proteasomal shuttle proteins (DDI1 and DDI2) is important to allow replication fork restart to occur (*Kottemann et al., 2018*). Additionally, the nuclear receptor interacting protein 3 (NIRP3) has been shown to upregulate DDI1 and increase polyubiquitylation of *Hs*RTF2, promoting *Hs*RTF2 removal and replication fork restart upon replication stress (*Suo et al., 2020*). Within disease models, *Hs*RTF2 has also been identified as a causal factor for Alzheimer's Disease, although the exact mechanism remains unclear (*Ou et al., 2021*; *Wingo et al., 2021*). To further elucidate the conserved function of Rtf2 at stalled replication forks, we investigated this protein further using a previously described *RTS1*-RFB system in *S. pombe* (*Naiman et al., 2021*).

In agreement with previous studies on Rtf2 at the *RTS1* RFB, we observe reduced barrier activity upon *rtf2* deletion as assayed both by polymerase-usage sequencing (Pu-seq) and a replication slippage assay. However, in our system, the mechanism of action of Rtf2 at *RTS1* does not occur via region A of *RTS1*, as had been previously reported. Using a proximity-based mass spectrometry method, we identified Rtf2 to be physically associated with mRNA processing and splicing factors. cDNA-Seq of mature polyA-mRNA revealed a modest global increase in the levels of intron retention in *rtf2Δ* cells. Intron retention was not uniform and specifically affected a subset of introns, with one of the most affected introns residing within the *rtf1* transcript. Using an intronless *rtf1* (*rtf1Δint*), that does not require splicing to encode a functional Rtf1 protein, we find that, in the absence of Rtf2, there was no reduction in RFB activity in comparison to when Rtf2 is present. These data demonstrate that the presence of Rtf2 is essential for the correct splicing of the *rtf1* transcript in order to allow optimal barrier activity at *RTS1*.

## Results

### Rtf2 is important for efficient barrier activity at RTS1

We have previously shown that RDR initiated at *RTS1* results in the formation of a non-canonical replication fork where DNA is semi-conservatively replicated with polymerase delta synthesising both the

leading and the lagging strands (*Miyabe et al., 2015*). Interestingly, the resulting replication is error prone (*Mizuno et al., 2013*), showing elevated replication slippage events (*Iraqui et al., 2012*). The non-canonical nature of the RDR fork provides us with two tools to follow replication restart: first, we can estimate the percentage usage of non-canonical replication from the levels of replication slippage measured using a genetic reporter that reconstitutes uracil prototrophy (*Iraqui et al., 2012*). Second, we can track DNA polymerase movement, and thus non-canonical RDR forks, using a recently developed Polymerase-usage sequencing (Pu-seq) method (*Naiman et al., 2021*). In brief, Pu-seq utilises pairs of mutant *S. pombe* strains, each harbouring an rNTP permissive mutation in the catalytic subunit of either of the main replicative polymerases (Polε or Polδ; *Keszthelyi et al., 2015*). In a ribonucleotide excision repair deficient background this allows rNMPs to persist in the DNA strand that is replicated by the mutant polymerase and enables subsequent strand-specific mapping of replication by each of the main replicative DNA polymerases (*Daigaku et al., 2015*).

In order to use Pu-seq to study the HR-restarted replication fork with minimal interference from converging canonical replication forks, the *RTS1* barrier sequence has been placed in a region of the genome next to an early firing origin with a distant late firing origin downstream (*Naiman et al., 2021*; *Figure 1A*). Thus, the predominant orientation of canonical replication across this region is in a rightward direction. The *RTS1* sequence has been inserted in the orientation that arrests replication forks originating from the early firing origin, but which is permissive to those originating from the late firing origin. In addition, to delay any replication forks initiating downstream of *RTS1* and thus increase the time available for replication forks blocked at *RTS1* to restart and replicate the downstream region, we inserted, 10 x ribosomal DNA RFB (rRFB) sequences (*Ter2-Ter3*) ~10 Kb downstream of *RTS1*. The rRFB is also polar and we have inserted the 10 x array in the orientation that will pause RFs originating from the late firing origin. The rRFBs do not collapse replications forks (*Mizuno et al., 2013*), do not utilise homologous recombination (HR) for restart (*Calzada et al., 2005*) and do not result in non-canonical replication forks that synthesise both strands with Polδ (*Naiman et al., 2021*). To allow comparison of HR-restarted replication forks to unhindered canonical replication forks, the *RTS1* RFB activity is controlled by the presence/absence of Rtf1 (*rtf1*[+]+ON, *rtf1Δ*=OFF).

Polymerase-usage Sequencing (Pu-seq) was first performed on strains containing the *RTS1* construct with either the barrier activity OFF (*rtf1Δ*) and barrier activity ON (*rtf1*[+]) (*Figure 1B*). When *RTS1* is OFF (*rtf1Δ*) the *RTS1* sequence and the ~10 kb to the right is replicated by Polε on the top strand and by Polδ on the bottom strand. This is consistent with predominant replication by rightward moving canonical forks. When the *RTS1* barrier is ON (*rtf1*[+]), there is an abrupt switch to Polδ usage on the top strand at the point at which *RTS1* has been inserted and the downstream region is replicated with both strands being synthesised by Polδ. This confirms that RDR is initiated at *RTS1* in most cells in the population, with both strands largely being replicated by a non-canonical fork across a region of ~10 kb.

As discussed above, Rtf2 has been reported to enhance the fork arrest at active *RTS1* when replication structures are visualised by 2D gels (*Codlin and Dalgaard, 2003*). To investigate how Rtf2 affects the dynamics of polymerase usage downstream of the *RTS1* RFB, Pu-seq was next conducted in *rtf2Δ* cells. As expected, canonical replication is evident across *RTS1* and the downstream region when the barrier is OFF (*rtf1Δ*, *rtf2Δ*) (*Figure 1C*). When *RTS1* is ON (*rtf1*[+] *rtf2Δ*), the switch from Polε usage to Polδ usage on the top is reduced at the site of *RTS1* when compared to *rtf2*[+] cells and Polδ usage on the top (leading) strand across the ~10 kb downstream region is also reduced. This indicates that, when compared to *rtf2*[+], a significantly reduced proportion of cells in the *rtf2Δ* population arrest the replication fork and switch to RDR to replicate the region downstream of the barrier (*Figure 1C*). This is particularly evident when the ratio of polymerase usage across both strands is calculated (*Figure 1D*). For canonical replication this is calculated to be Polδ:Polε=50:50, whereas with increasing levels of RDR in the population the same calculations are expected to generate a bias towards Polδ:Polε=100:0. In *rtf2*[+] cells, there is a clear Polymerase δ bias at and downstream of active *RTS1* (ON: *rtf2*[+]; *Figure 1D*). However, when *rtf2* is deleted the level of Polδ bias produced at and downstream of the *RTS1* barrier sequence is reduced. As expected, in the absence of *rtf1* (OFF: *rtf1Δ*) the ratio of Polδ:Polε=approximately 50:50, irrespective of the presence or absence of *rtf2*.

We further confirmed these findings by measuring the level of mutagenesis at a region potentially replicated by the restarted replication fork. To assay mutagenesis the *ura4-sd20* allele, which contains a 20 bp tandem repeat and renders cells uracil dependent (*Iraqui et al., 2012*), was integrated

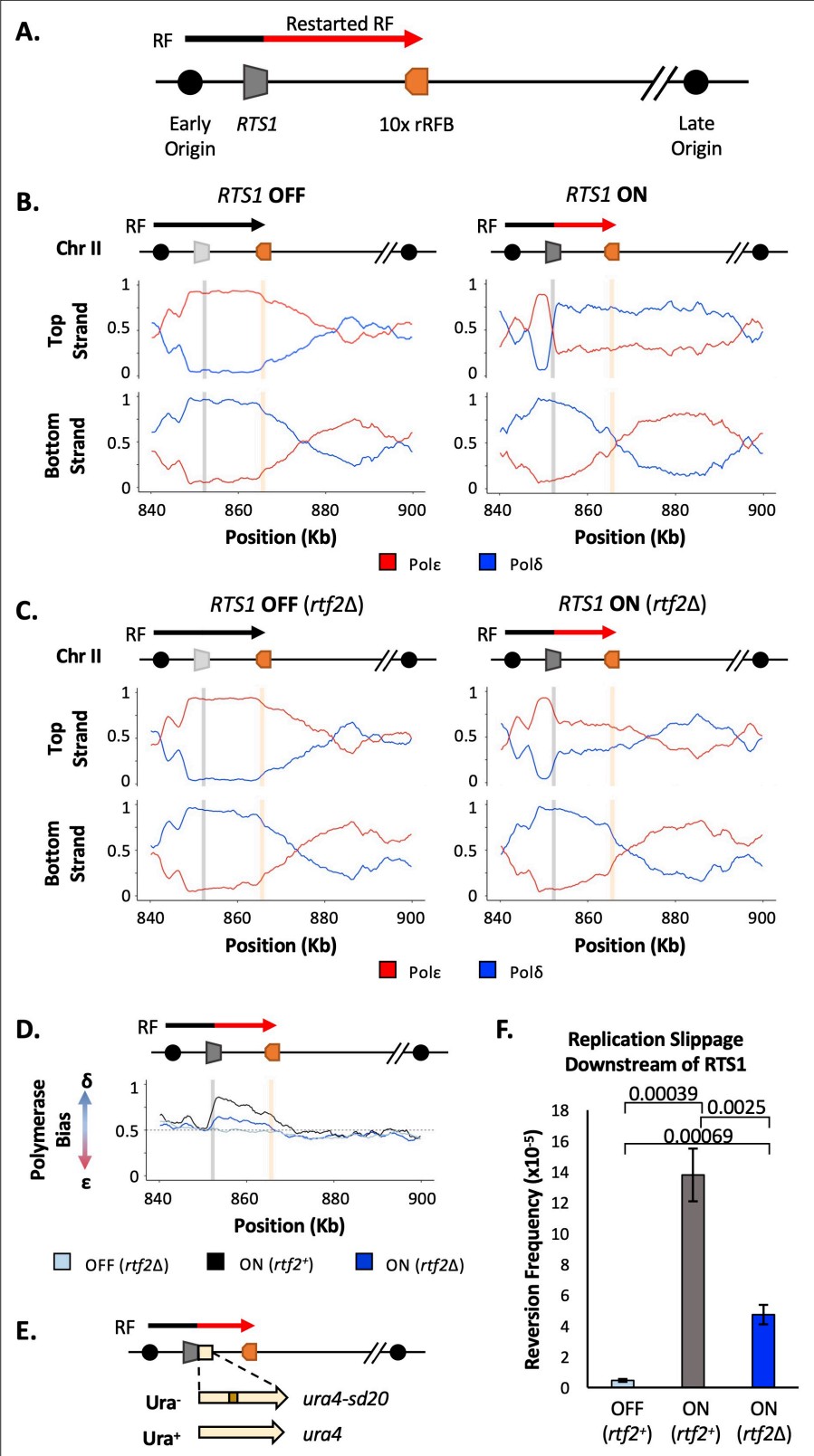

**Figure 1.** Rtf2 deletion reduces replication fork restart at *RTS1*. (**A**) The *RTS1* sequence (grey box) is inserted between an early and a late firing replication origin and 10 ribosomal replication fork barriers (10 x rRFB, orange box) are inserted ~10 Kb downstream of *RTS1*. The predominant direction of replication is shown with canonical (black) and restarted replication forks (red) indicated. (**B**) Polymerase usage around the *RTS1* RFB locus in *rtf2*⁺

*Figure 1 continued on next page*

*Figure 1 continued*

cells. *RTS1* OFF (left panel) and ON (right panel). The ratio of Polymerase ε (red) and Polymerase δ (blue) for both the top and bottom strand is shown. (**C**) Polymerase usage around the *RTS1* RFB locus in *rtf2Δ* cells. *RTS1* OFF: rtf1Δ rtf2Δ. *RTS1* ON: *rtf1⁺ rtf2Δ*. Panel details as in B. (**D**) Polymerase bias graph calculated using the ratio of polymerase usage across both strands around the *RTS1* RFB locus. (**E**) Schematic of the *RTS1*-RFB replication slippage assay. A *ura4* allele containing a 20 bp tandem repeat (*ura4-sd20*) is inserted immediately downstream of the *RTS1* sequence. Replication slippage can result in loss of one repeat, which manifests as ura⁺. (**F**) Replication fork slippage events scored as the frequency of *ura4⁺* reversions over 2 cell cycles using the *RTS1-ura4-sd20* replication fork slippage assay. Data from three independent experiments ± SD. Statistical analysis by two-tailed Students T-test.

immediately downstream of the *RTS1* barrier (***Figure 1E***). Deletion of one tandem repeat by replication slippage will revert the allele to *ura4⁺*, resulting in uracil prototrophic cells. These events can subsequently be selected for, and quantified, as a readout for replication fork slippage events. In *rtf2⁺* cells, the activation of *RTS1* results in a large increase in replication fork slippage events in the downstream region when compared to cells where *RTS1* is OFF (***Figure 1F***). For *rtf2Δ* cells, there is a reduction in replication fork slippage events downstream of an active *RTS1* in comparison to *rtf2⁺* cells, but this was not reduced to the low levels seen when *RTS1* was OFF. This reduction in replication fork slippage when *rtf2* is deleted is fully consistent with, and likely reflects, the reduced levels of restarted replication forks evident in the Pu-seq traces.

Taken together, these results suggest that an increased proportion of replication forks remain canonical (ε/δ) after encountering the *RTS1* RFB in the absence of Rtf2, with only around a third of replication forks (based on the proportion of polymerase bias of *rtf2Δ* compared to *rtf2⁺*) arresting and restarting using non-canonical (δ/δ) replication. Therefore, in *rtf2Δ* cells, only a subset of replication forks block at *RTS1* and restart via RDR when compared to the *rft1⁺*. The other replication forks either arrest briefly at *RTS1* and resume as a canonical replication fork, or they do not get blocked at *RTS1* and instead continue replication across the region as in an *RTS1* OFF situation.

## Rtf2 does not increase RTS1 barrier activity via interactions with enhancer region A

*RTS1* is annotated as being composed of two main regions, A and B (***Figure 2A***). Region B contains four repetitive sequence motifs and is proposed to bind Rtf1 (***Codlin and Dalgaard, 2003***). It has previously been reported that Rtf2 enhances *RTS1* barrier activity by either direct or indirect association with region A. This was because a similar and non-additive reduction in the blocking signal (monitored by 2D gel electrophoresis) was observed for a plasmid-borne *RTS1* construct lacking region A as was evident in *rtf2Δ* cells (***Codlin and Dalgaard, 2003***). To confirm this observation in our *RTS1* system and characterise any impact of region A on polymerase usage downstream of an active (*rtf1⁺*) *RTS1*, region A was deleted from the *RTS1* sequence and the modified *RTS1_AΔ* construct inserted to replace *RTS1*. The modified locus still contains the downstream 10xrRFBs. The truncated *RTS1_AΔ* sequence was also incorporated into the replication fork slippage construct, to produce *RTS1_AΔ:ura4-sd20*. If Rtf2 is performing an enhancing function for arrest at *RTS1* via region A, the same profiles of mutagenesis and polymerase usage would be expected for *rtf2Δ* and *RTS1_AΔ*.

We first monitored the levels of mutagenesis downstream of *RTS1* using *RTS1:ura4-sd20* and *RTS1_AΔ:ura4-sd20* (***Figure 2B***) loci. Similar basal levels of replication fork slippage were seen in both systems when *RTS1* was OFF (*rtf1Δ*). As expected, replication fork slippage rates were also similar when *rtf2* is deleted in both systems when the barrier is ON (*rtf1⁺rtf2Δ*). If region A is required for the enhancing effect on fork arrest at *RTS1* by Rtf2, the levels of replication fork slippage when the barrier is ON would be expected to be reduced in *RTS1_AΔ rtf2⁺* cells to the same level as seen in *RTS1 rtf2Δ*. However, this was not the case and equivalent levels of replication fork slippage are observed for both *RTS1* and *RTS1_AΔ* constructs in the ON (*rtf1⁺ rtf2⁺*) state. This suggests that region A is dispensable for *RTS1* arrest and RDR restart efficiency in our system and is not likely to be a site of Rtf2 function.

To establish the extent of RDR occurring in the absence of region A, Pu-seq was performed on strains containing the *RTS1_AΔ* system. As expected, when barrier activity was OFF (*rtf1Δ*) replication was canonical across the region, as has previously been seen for *RTS1* (***Figure 2C***; top). When the barrier was ON (*rtf1⁺*), the same levels of Polδ bias (representing non-canonical RDR) was observed for

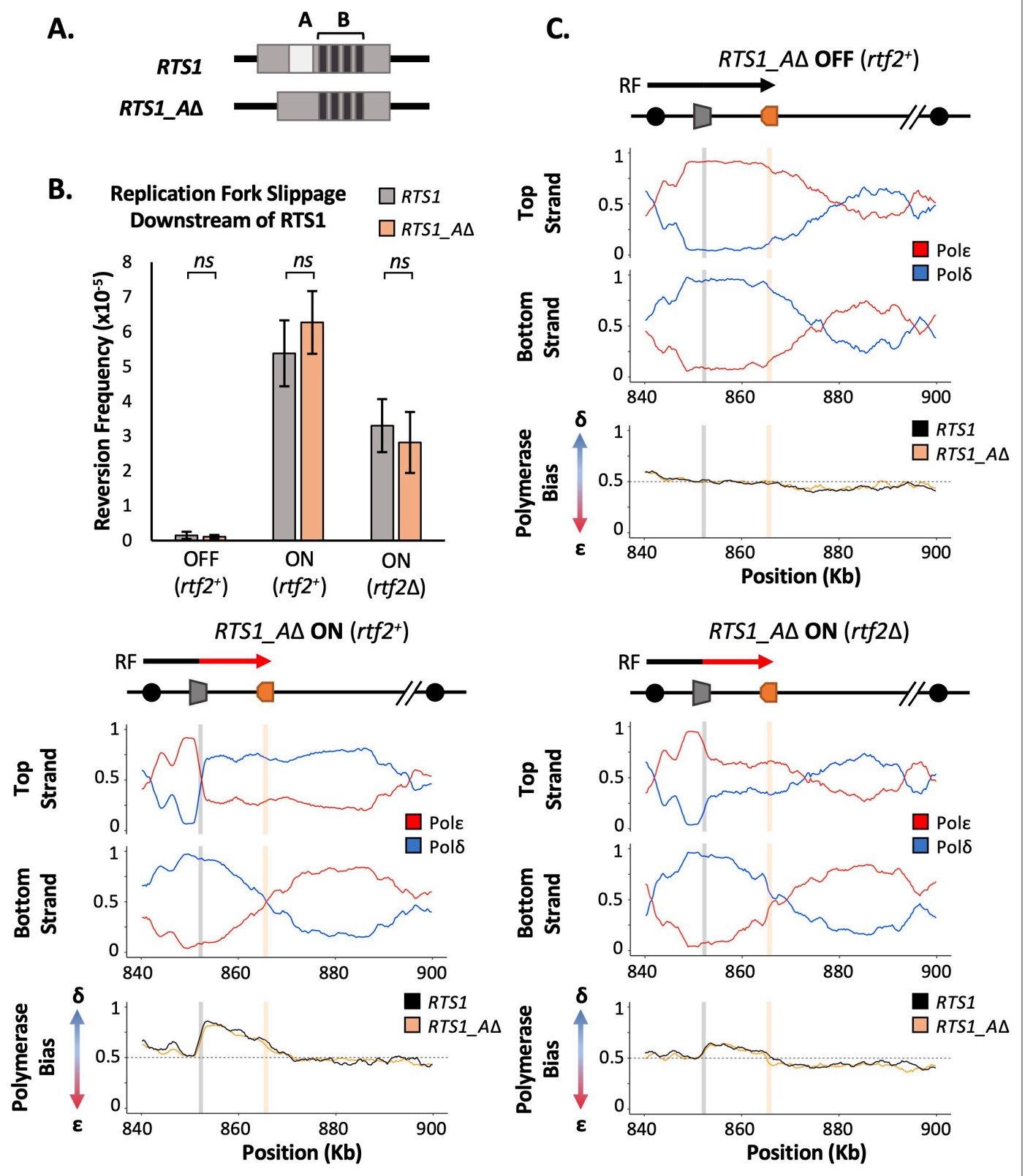

**Figure 2.** *RTS1* region A is dispensable for efficient replication fork restart. (**A**) Schematic of the *RTS1* RFB. Region A is a ~60 bp purine rich region and Region B is ~450 bp containing the four repeated sequence motifs essential for *RTS1* activity. (**B**) Replication fork slippage events scored as the frequency of *ura4*⁺ reversions over two cell cycles. Data from three independent experiments ± SD. Statistical analysis by two-tailed Students T-test, p>0.05 = not significant (*ns*). (**C**) Polymerase usage around the *RTS1_A*Δ RFB locus for RFB OFF (Top panel) and ON (bottom left panel) for *rtf2*⁺ and

*Figure 2 continued*

polymerase usage around the *RTS1_A*Δ RFB locus in *rtf2*Δ cells with the barrier ON (bottom right panel). For each panel the ratio of Polymerase ε (red) and Polymerase δ (blue) for both the top and bottom strand is shown along with a polymerase bias graph calculated using the ratio of polymerase usage across both strands and comparing *RTS1_A*Δ (orange) with the *RTS1* locus (black).

The online version of this article includes the following source data and figure supplement(s) for figure 2:

**Figure supplement 1.** Mud1 does not impact the turnover of Rtf2 or cell sensitivity to HU/MMS.

**Figure supplement 1—source data 1.** Source data for *Figure 2—figure supplement 1*.

**Figure supplement 2.** Uncropped blots for *Figure 2—figure supplement 1*.

*RTS1_A*Δ as was seen for *RTS1* (*Figure 2C*; bottom left). Similarly, when the barrier activity was ON and Rtf2 function was absent (*rtf1⁺*, *rtf2*Δ) the same reduced level of Polδ bias was observed for both *RTS1_A*Δ and *RTS1* (*Figure 2C*; bottom right). Taken together, these results indicate that Rtf2 does not specifically interact with, or function through, region A and that region A is dispensable for normal levels of fork arrest and restart by RDR at this locus.

## DDI1 homolog in *S. pombe* (Mud1) is not responsible for the degradation Of Rtf2

Having shown that Rtf2 is not interacting with *RTS1*, we aimed to determine if Rtf2 was playing a similar role through *RTS1* as had been suggested for its role at stalled replication forks in human cells (*Kottemann et al., 2018*; *Suo et al., 2020*). Human RTF2 is removed from stalled replication forks and targeted to the proteasome via proteasomal shuttle proteins DDI1/2. To establish if *S. pombe* Rft2 is a client protein for the equivalent proteasomal shuttle, we deleted the *S. pombe* gene encoding the DDI1/2 homolog, *mud1* (*Trempe et al., 2005*), and determined cell sensitivity to agents capable of stalling replication forks (*Figure 2—figure supplement 1A*). DDI1/2 knockdown in human cells sensitises to HU treatment, and *S. pombe rtf2*Δ cells are sensitive to high levels of MMS. Deletion of *mud1* did not result in sensitivity to MMS (*Figure 2—figure supplement 1A*) and a double deletion mutant, *mud1*Δ *rtf2*Δ, displayed no increase in MMS sensitivity in comparison to *rtf2*Δ alone. Similarly, no sensitivity to HU was observed for *mud1*Δ, *rtf2*Δ or the double *mud1*Δ *rtf2*Δ. These results suggest that Mud1 does not play the same role as has been observed for human DDI1/2 proteins.

To further investigate the equivalence between Mud1 and human DDI1/2, we monitored the turnover of Rtf2 in the presence and absence of Mud1. Upon treatment with cycloheximide, a translation inhibitor, the levels of Rtf2 are rapidly decreased (*Figure 2—figure supplement 1B*). In a *mud1*Δ background, the rate of Rtf2 degradation was equivalent to *mud1⁺* +. This further suggests that Mud1 is not playing the same role in regulating Rtf2 in *S. pombe* as has been observed for DDI1/2 in human cells.

## Rtf2 is associated with mRNA processing and splicing factors

Having established that Rtf2 is not enacting its role on the levels of replication fork restart at *RTS1* via interaction with region A, we sought to investigate if it is instead travelling with the replication fork, as has been suggested for human RTF2 (*Kottemann et al., 2018*). A proximity biotin labelling-based mass spectrometry method, TurboID (*Branon et al., 2018*), was used to identify associated proteins. TurboID utilises a mutant version of the *E. coli* BirA biotin ligase (TbID) that catalyses biotin into biotinoyl-5'-AMP, which subsequently covalently attaches to proteins within close proximity (*Choi-Rhee et al., 2004*). Rtf2 was C-terminally tagged with BirA^TbID and an internal 3 HA tag. To establish if the tag negated Rtf2 function, a spot test was conducted to monitor sensitivity to high levels of MMS (*Figure 3—figure supplement 1A*). There was no evident sensitivity of *rtf2-3HA-TurboID* cells, indicating the tag did not negate Rtf2 function.

*rtf2-3HA-TurboID* and control *rtf2⁺* cells were synchronised in S phase to enrich for replication (*Figure 3—figure supplement 1B*) structures or treated with MMS to enrich for stalled/collapsed replication forks. Following biotin addition, purification of biotin tagged proteins by streptavidin affinity and mass spectrometry, proteins that were enriched with statistical significance in comparison to no TurboID tag were identified for both conditions (*Figure 3—figure supplement 1C* and *Supplementary file 1*). As expected, Rtf2 was identified as a hit with a large increase in abundance. A number of additional proteins were also identified as being significantly enriched in the *rtf2-3HA-TurboID*

cells, and thus predicted to be in close proximity to the Rtf2[TbID] (*Figure 3—figure supplement 1C*). To determine which processes Rtf2 may be involved in, GO Term analysis was conducted on all significant hits for each condition (*Figure 3—figure supplement 1D* and *Supplementary files 2 and 3*). Based on the identification of human RTF2 as a replication factor, GO Terms associated with replication and associated processes were expected to be identified. Surprisingly, no such GO Terms were associated with the proteins identified as being in close proximity to Rtf2. Instead, the top GO Terms that arise are those to do with mRNA processing and splicing. This supports previous mass spectrometry results for *Arabadopsis Thaliana* Rtf2 (*At*Rtf2), which similarly identified splicing factors as Rtf2 interactors (*Sasaki et al., 2015*). Additionally, this is in concordance with a genome wide screen in *S. pombe* that identified Rtf2 as affecting splicing of two reporter introns (*Larson et al., 2016*).

## Rtf2 deletion results in increased intron retention

To test if Rtf2 has an effect on splicing patterns, long-read direct cDNA-sequencing was conducted using an Oxford Nanopore MinION sequencer (*Jain et al., 2015*). Differential splicing can be efficiently identified by sequencing full length cDNA created from polyA-mRNA transcripts (*Bolisetty et al., 2015*). The polyA tail is only added to mature mRNA allowing the polyA tail to be used for purification of only those transcripts that have completed splicing. Sequencing was carried out on cDNA derived from polyA-mRNA samples from *rtf2*+ and *rtf2*Δ cells to get an overview of alterations to splicing patterns across entire transcripts. Sequencing reads were mapped to the genome using minimap2 (*Li, 2018*) and mapped transcripts were compared to the annotated transcripts using GffCompare (*Pertea and Pertea, 2020*), which reports statistics (see *Figure 3A* right and *Supplementary file 4*) relating to the measure of agreement of the input transcripts when compared to reference annotation.

The proportion of reads for each type of mapped transcript did not vary significantly between *rtf1*+ and *rtf2*Δ, except for those indicating an intron retention event (type 'm') (*Figure 3A*). Having observed an increase in intron retention at the transcript level, we sought to confirm this via quantification of retention for each individual intron. Quantifying the fraction of mapped reads that span each intron vs. those that lack the intron and only map to the two flanking exons confirmed a shift toward increased intron retention in *rtf2*Δ cells (*Figure 3B* and *Supplementary file 5*). Deletion of *rtf2* increases intron retention in only a subset of total introns, with 43 genes consistently resulting in transcripts with an intron retention event occurring 20% more often in at least 2 of three repeats in comparison to *rtf2*+ (*Supplementary file 6*). However, this number is likely an underestimate due to reduced read depth of low expression genes as well as degradation to the 5' end of some transcripts.

Of the increased intron retention transcripts, the *RTS1* DNA binding factor *rtf1* was identified, whose activity is crucial for the barrier activity (*Figure 3C*). Visual inspection of the transcript map produced from GffCompare clearly shows intron 2, which is usually an efficiently spliced intron, persists in essentially all the transcripts sequenced (type: 'm'). A second *rtf1* intron, intron 6, is also affected, showing retention in a proportion of the transcripts. The GffCompare transcript map shows that *rtf1* transcripts from *rtf2*+ cells splice all introns effectively: i.e., are classed as equivalent to the reference genome (type: '=') and a single predominant transcript is predicted. However, two predominant *rtf1* transcripts were predicted by GffCompare for *rtf2*Δ cells, both of which retain intron 2 and one of which also retained intron 6 (type: 'm'). To confirm these results, PCR amplification of each of these introns from cDNA derived from mature polyA-mRNA was performed (*Figure 3D*). Upper bands indicate the intron-retained isoform, while the lower bands indicate the correctly spliced isoform. For intron 2, the spliced isoform is undetectable, consistent with the read coverage from the cDNA-Seq data. There is a modest increase in the retained isoform for intron 6 in *rtf2*Δ in comparison to *rtf1*+. These data are fully consistent with the role of Rtf2 at *RTS1* being indirect, having its effect via ensuring the correct splicing of *rtf1*.

The subset of genes identified as having introns retained in *rtf2*Δ cells were analysed further. There were no obvious changes to overall gene expression or expression of 'intron-retained' (GffCompare group 'm') genes when analysing the abundance of transcripts between *rtf2*Δ and *rtf1*+ (*Figure 3—figure supplement 2A*). The length of intron did not correlate with changes to intron retention (*Figure 3—figure supplement 2B*). The only correlation identified was a decrease in GC richness across the branch point (BP), 3' splice site (3'SS) and the first ~100 bp of the downstream exon in those introns with the highest rates of intron retention (*Figure 3—figure supplement 2C*). This reduction in GC richness fell outside of the 95% confidence interval calculated for a sample of 100 intron:exon

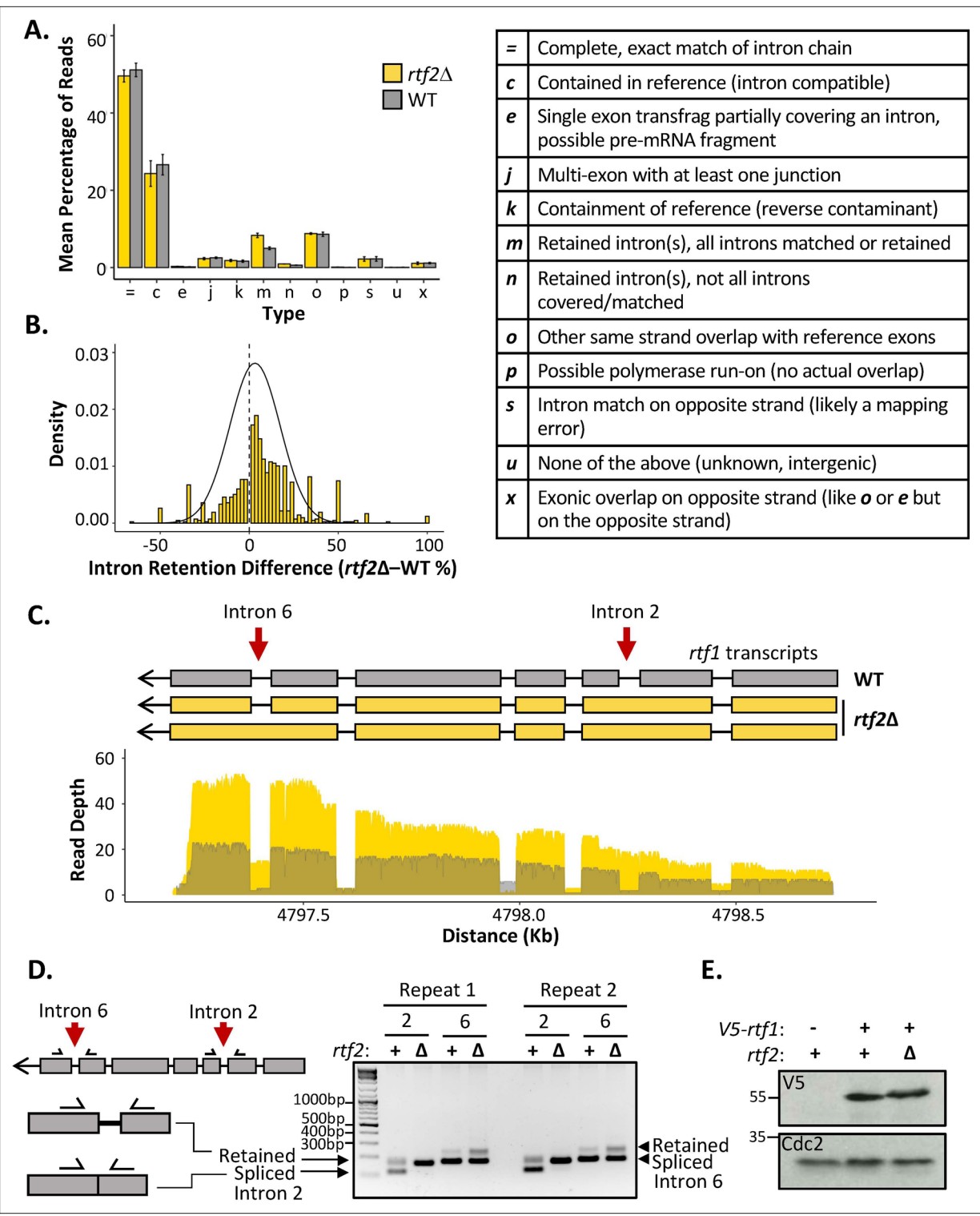

**Figure 3.** *rtf2Δ* results in increased retention of a subset of introns including within the *rtf1* transcript. (**A**) GffCompare classification analysis of transcripts from WT (grey) and *rtf2Δ* (yellow) samples. Graph shows the mean of three biological repeats with error bars representing SD. The table (right) describes each classification. (**B**) Graph showing the difference in intron retention for each individual intron with a normal distribution curve fitted. Introns that show no difference between WT and *rtf2Δ* are not included on the graph. (**C**) Depth of reads mapped across the *rtf1* transcript for WT (grey) and *rtf2Δ* (yellow) samples. Corresponding transcripts as calculated by GffCompare are shown above. (**D**) Two repeats of PCR amplification of intron 2 and intron 6 from cDNA derived from polyA-mRNA. Schematic shows the principle behind the shift in size from a smaller band, representing correctly spliced,

*Figure 3 continued on next page*

*Figure 3 continued*

to a larger band, representing a retained intron. (**E**) Whole cell extract of *V5-rtf1* containing cells in WT and *rtf2Δ* background probed with indicated antibody. Cdc2 is shown as a loading control.

The online version of this article includes the following source data and figure supplement(s) for figure 3:

**Source data 1.** Source data for *Figure 3*.

**Figure supplement 1.** Proximal Proteins of Rtf2 identified via TurboID Proximity Based Labelling Mass Spectrometry.

**Figure supplement 2.** cDNA-Seq analysis of retained introns.

**Figure supplement 3.** Uncropped blots for *Figure 3*.

sequences indicating this decreased GC content as a significant pre-requisite for the increased intron retention in *rtf2Δ* cells.

## Intronless rtf1 rescues the rtf2Δ phenotype restoring full RTS1 RFB activity

Intron 2 of the *rtf1* gene is 45 bp and, if not spliced, the adjacent exons remain in frame and a 15 amino acid insertion is predicted to occur between amino acids 202–203. To establish how the intron retention of *rtf1* in *rtf2Δ* cells effects the final protein product, we N-terminally tagged *rtf1* with a V5 tag and performed a western blot on a whole cell extract (*Figure 3E*). The protein levels of Rtf1 do not vary between *rtf2+* and *rtf2Δ* cells. However, there is a small upward shift in the migration of the protein in *rtf2Δ* cells, which likely corresponds to the small increase in protein size expected from intron 2 retention.

Rtf2 clearly plays an important role in the correct splicing of *rtf1* mRNA. Incorrect splicing of the *rtf1* transcript could reduce the functionality of the protein and result in reduced binding to *RTS1* or the inefficient blocking of replication forks at this sequence. To test if the increased intron retention in *rtf1* mRNA is indeed responsible for reduced *RTS1* activity in *rtf2Δ* cells, we replaced the wildtype *rtf1* gene with an intronless *rtf1* gene (*rtf1Δint*) at its native locus (*Figure 4A*). When transcribed, *rtf1Δint* can no longer retain an intron and thus will mimic the largely efficient splicing that occurs in a *rtf2+* background (c.f. *Figure 3C*). To establish if *rtf1Δint* rescues the defect at *RTS1* RFB in *rtf2Δ* cells we first tested this allele in the replication fork slippage assay (*Figure 4B*). When *RTS1* is ON, both *rtf1+* and intronless *rtf1Δint* exhibit the same levels of replication fork slippage downstream of *RTS1*, indicating similar barrier functionality. However, when *rtf2* is deleted in the presence of *rtf1Δint*, fork slippage levels do not drop to the level seen in *rtf2Δ rtf1+* cells, and instead remains at the higher frequency seen for *rtf2+ RTS1* ON. This increase in mutagenicity downstream of *RTS1* for *rtf1Δint* in *rtf2Δ* cells indicates the *RTS1* RFB activity is restored to *rtf2+* levels in the absence of Rtf2.

To confirm the increased mutagenicity downstream of *RTS1* in *rtf2Δ rtf1Δint* cells is due to *rtf2+* levels of RFB activity and replication fork restart, *rtf1Δint* was also analysed by Pu-seq (*Figure 4C*). In both *rtf2+* and *rtf2Δ* backgrounds, the presence of *rtf1Δint* resulted in the same high levels of Polδ bias downstream of active *RTS1*, which are equivalent to levels seen for *rtf1+*, *rtf2+* *+TS1* ON (c.f. *Figure 1B*). Raw Pu-seq traces also clearly show Polδ to be the predominant polymerase used in both the top and bottom strands downstream of *RTS1* (*Figure 4—figure supplement 1*). This demonstrates that the presence of Rtf2 is essential for the correct splicing of Rtf1 to allow efficient barrier activity at *RTS1*.

## Discussion

The finding that the removal of human RTF2 from stalled replication forks is important to allow replication restart and maintain genome stability (*Kottemann et al., 2018*) led us to investigate the function of *S. pombe* Rtf2 in more detail. The Rtf2 protein was originally identified as having the role of enhancing the blocking capacity of the *RTS1* RFB in *S. pombe* (*Codlin and Dalgaard, 2003*). 2D gel analysis of replicating plasmids containing *RTS1* revealed a reduced pausing signal when cells were deleted for *rtf2*. A further study identified a visible increase in large Y-intermediates that were dependent on the Srs2 helicase (*Inagawa et al., 2009*). This was interpreted as Rtf2 acting downstream of Rtf1 to convert replication barrier activity at the *RTS1* locus into a replication termination site.

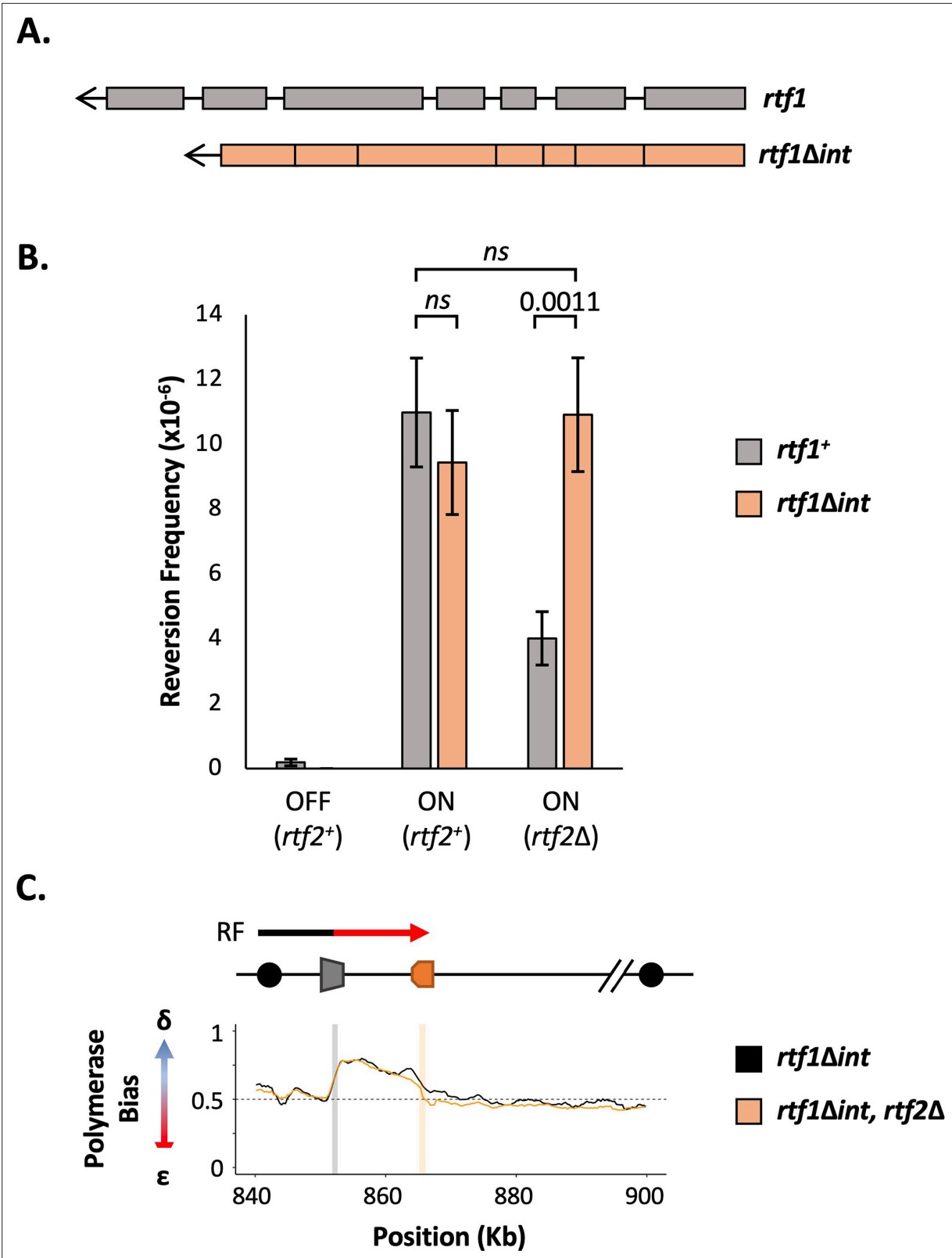

**Figure 4.** Intronless *rtf1* rescues the *rtf2Δ* phenotype restoring full *RTS1* RFB activity. (**A**) Representative schematic of the *rtf1+* gene containing both exons and introns, and the intronless version, *rtf1Δint* that contains only exons. *rtf1Δint* was inserted at the native locus and is under the control of the native promoter. (**B**) Replication fork slippage events scored as the frequency of *ura4+* reversions. Data from at least three independent experiments ±

*Figure 4 continued on next page*

*Figure 4 continued*

SD. Statistical analysis was by two-tailed Students T-test, p>0.05 = not significant (*ns*). (**C**) Polymerase bias graph calculated using the ratio of polymerase usage across both strands at the *RTS1* RFB In cells expressing *rtf1Δint* and either with or deleted for *rtf2*⁺.

The online version of this article includes the following source data and figure supplement(s) for figure 4:

**Figure supplement 1.** Intronless *rtf1* raw Pu-seq traces.

**Figure supplement 2.** Rtf1 structural predictions.

**Figure supplement 2—source data 1.** Source data for *Figure 4—figure supplement 2*.

**Figure supplement 3.** Rtf1 with the amino acid sequence from a non-excisable intron 2 does not increase slippage.

**Figure supplement 4.** Uncropped blots for *Figure 4—figure supplement 3*.

Here, we confirm that the loss of Rtf2 indeed results in a decrease in replication fork arrest and restart by RDR at the *RTS1* RFB. However, we clearly identify Rtf2 as interacting with splicing factors and demonstrate increased intron retention in a subset of transcripts when *rtf2* is deleted. Importantly, we found that intron 2 of Rft1, a Myb-domain binding protein required for *RTS1* barrier activity (*Eydmann et al., 2008*), is not spliced in *rtf2Δ* cells and this results in an Rtf1 protein that contains a 15 amino acid insertion between residues 202–203. Rtf1 contains 5 domains that resemble Myb/SANT domains. A DALI search of an AlphaFold2 model of Rtf1 indicates a high degree of similarity with the crystal structure of *S. pombe* Reb1, which promotes replication pausing within the rDNA spacer region (*Figure 4—figure supplement 2*). The 15 amino acid insertion is unlikely to disrupt DNA binding directly, but appears to extend a helix that, based on our observations, renders the resulting Rtf1 protein dysfunctional (*Figure 5*). Replacing the genomic *rtf1* gene with an intronless copy fully rescued the *rft2Δ*-dependent defect in fork arrest at *RTS1*, clearly demonstrating that Rft2 acts upstream of Rtf1 to allow the correct splicing of the *rft1* mRNA and thus the production of fully functional Rtf1 protein. We also confirmed that expression of an intron-less *rtf1* gene that encodes the same additional 15 amino acids between residues 202–203 (Rtf1 intron2NE) does not provoke increased replication slippage downstream of *RTS1* (*Figure 4—figure supplement 3*), indicating that this protein is indeed dysfunctional.

Using 2D gel analysis of plasmids containing various deletion constructs of *RTS1*, previous work suggested Rtf2-dependent enhanced blocking capacity at *RTS1* functions via an interaction with Region A of the *RTS1* sequence (*Codlin and Dalgaard, 2003*). In our genomic *RTS1* system, we showed that deletion of Region A (*RTS1_AΔ*), does not affect the levels of non-canonical (δ/δ) replication forks or the levels of replication fork slippage downstream of active *RTS1* (*Figure 2*). Therefore, these results contrast with the previous finding and demonstrate that region A of *RTS1* is dispensable for the efficiency of barrier activity and not to be the site of Rtf2 interaction.

RTF2 in human cells has been shown to be enriched at nascent chromatin (*Dungrawala et al., 2015*; *Kottemann et al., 2018*). However, we see no evidence to suggest that *S. pombe* Rtf2 is associated with the replication fork from our biotin proximity labelling experiments: Mass spectrometry analysis of proteins within close proximity to Rtf2 did not identify any replication associated factors (*Figure 3—figure supplement 1*). The studies in human cells specifically enriched for nascent chromatin and it is therefore possible that this allowed higher sensitivity for Rtf2 enrichment when compared to our TurboID experiments. Of note, a previous study *Inagawa et al., 2009* found Rtf2 to co-precipitate with *S. pombe* Pcn1 (PCNA). We did not find any evidence that Pcn1 was enriched in our TurboID MS data. It is possible that a Pcn1:Rtf2 interaction does occur, but is very transient and only visualised in the previous study due to the use of overexpression plasmids.

Rtf2 is an abundant protein that is predicted to contain a RING motif similar to that found in the E3 SUMO ligases Pli1 and Nse2 in *S. pombe* (*Watts et al., 2007*) and potentially related to RING domains of ubiquitin E3 ligases. Furthermore, deletion of *pmt3* (*S. pombe* SUMO) has been reported to result in a similar decrease in replication fork stalling at *RTS1* as *rtf2Δ* (*Inagawa et al., 2009*), leading to speculation that Rtf2 acts at the *RTS1* stall site to SUMO modify replication factors. However, it has also been shown that ubiquitylation and sumoylation of splicing factors are important to maintain efficient splicing within a cell (*Pozzi et al., 2017*) and therefore it is also conceivable that the efficient splicing of *rtf1* intron 2, as well as other introns, is dependent on either sumoylation and/or ubiquitylation (*Pozzi et al., 2018*) of splicing factors by Rtf2. The balance of evidence suggests that Rtf2 has evolved a new function(s) in human cells to modulate replication fork restart: we are unable

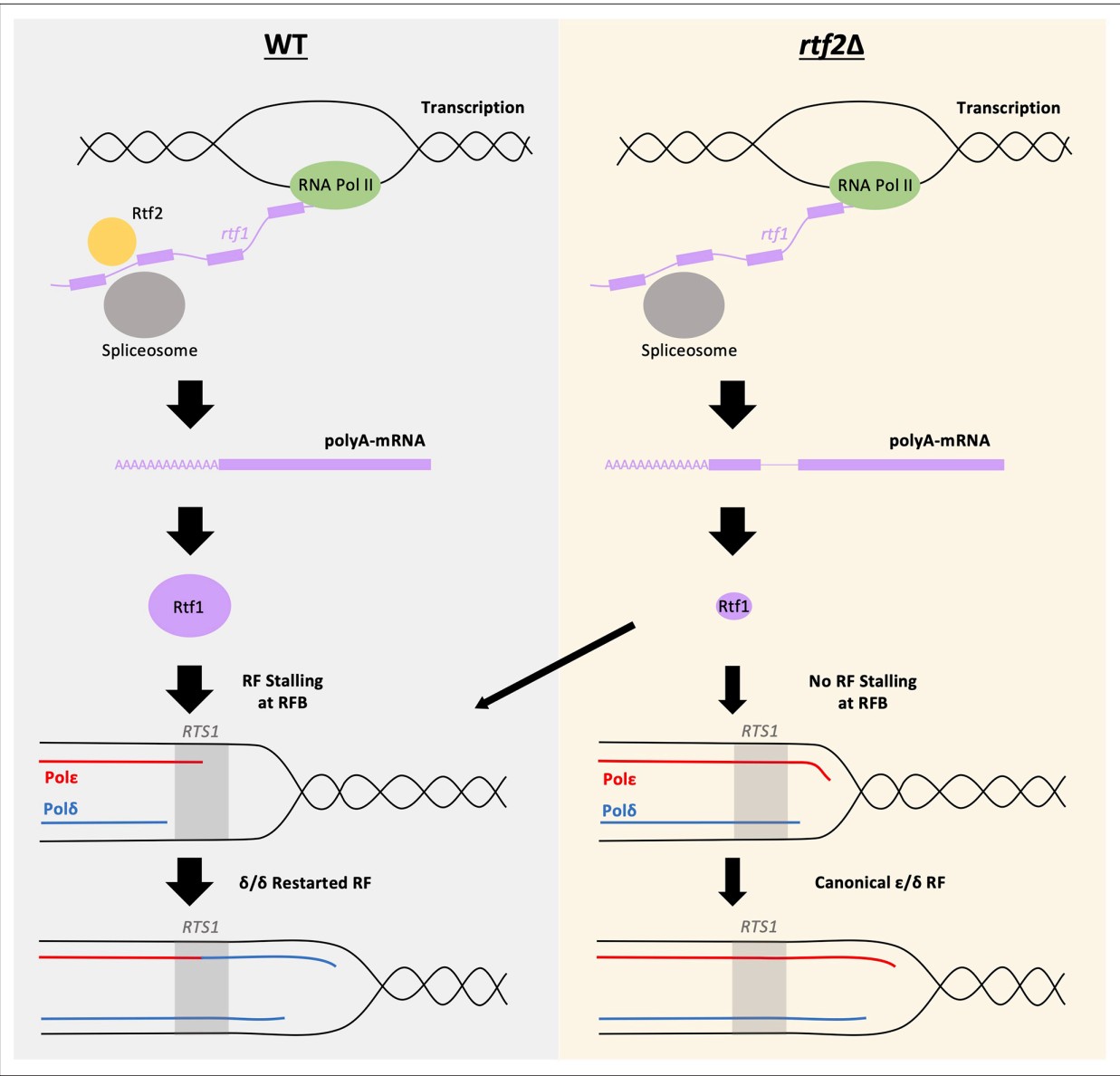

**Figure 5.** Mis-splicing of *rtf1* mRNA in *rtf2Δ* cells results in reduced *RTS1* RFB Activity. *Left panel*: In wildtype cells Rtf2 is present and allows for the correct splicing of *rtf1* mRNA and the production of functional Rtf1 protein. This results in replication fork stalling at *RTS1* and RDR, which produces non-canonical δ/δ replication forks. *Right panel*: In *rtf2Δ* cells, *rtf1* mRNA is mis-spliced resulting in retention of intron 2. This produces an Rtf1 protein less capable of blocking replication forks at *RTS1* manifesting as some forks able to bypass *RTS1* unhindered and continue as canonical replication forks. A small portion of forks are still arrested and restart by RDR resulting in a small proportion of non-canonical δ/δ replication.

to identify any evidence that Rtf2 associates with replication proteins, *rtf2* deletion does not directly affect *RTS1*-dependent fork arrest or subsequent restart by RDR; *rtf2* deletion cells are not sensitive to the replication inhibitor HU. Furthermore, in human cells RTF2 is actively degraded via the DDI1/2 proteasomal shuttle system to regulate its activity at stalled forks. In *S. pombe,* we saw no evidence that the DDI1/2 homolog, Mud1, is influencing Rtf2 stability or the response to HU treatment.

Rtf2 has also been identified in *Arabidopsis thaliana* as an essential protein that contains an additional non-conserved N-terminal extension (*Sasaki et al., 2015*). Mass spectrometry to identify *At*Rtf2 interacting proteins revealed proteins involved in mRNA splicing, RNA binding and metabolism, as well as DNA binding and ribosomal proteins. Mass spectrometry for proteins interacting with *At*Rtf2 truncated for the non-conserved N-terminal domain identified many of the same proteins, indicating the interactions and their specificity to the conserved core of RTF2 are likely to be conserved between *S. pombe* and humans. Furthermore, the cDNA-Seq we conducted identified a subset of introns that

were inefficiently spliced in *rtf2Δ* cells (*Figure 3—figure supplement 2*). This is consistent with the study in *A. thaliana*, which also detected intron retention defects when *AtRTF2* was deleted (*Sasaki et al., 2015*). Interestingly, a genome wide screen in *S. pombe* for factors affecting mRNA splicing similarly identified deletion of *rtf2* to result in intron retention defects (*Larochelle et al., 2019*), supporting our cDNA-Seq results (*Figure 3*). Whether Rtf2 acts directly as a splicing factor or indirectly effects spliceosome function remains to be elucidated.

Rtf2 is a conserved protein that affects splicing in fission yeast and plants and is likely to affect splicing in humans. In *S. pombe,* the major phenotype that has been observed relates to replication fork arrest at the *RTS1* barrier, but other biological processes (for example a cellular response to MMS in fission yeast - *Figure 2—figure supplement 1*) are likely to be affected by inefficient spicing of other proteins. It is also likely that other pathways are directly influenced by Rft2 function, as appears to be the case in human cells (*Kottemann et al., 2018*; *Suo et al., 2020*). Recently, Rtf2 has been implicated in a number of additional physiological functions and pathologies including restriction of viral infection (*Chia et al., 2020*) and a possible causal factor for Alzheimer's disease in humans (*Wingo et al., 2021*; *Ou et al., 2021*). Alzheimer's has previously been associated with differential splicing patterns (*Raj et al., 2018*) raising the possibility that the mis-splicing observed in Alzheimer's patients could be connected to changes in RTF2 protein function in these patients. This needs to be investigated further but could provide a better understanding of the disease's progression.

In summary, we identify *S. pombe* Rtf2 as a key factor for the efficient splicing of a subset of introns and demonstrate that this explains the effects of *rtf2Δ* at the site-specific replication fork barrier *RTS1* (*Figure 5*).

## Methods

### Key resources table

| Reagent type (species) or resource | Designation | Source or reference | Identifiers | Additional information |
|---|---|---|---|---|
| Antibody | Anti-HA (mouse monoclonal) | SantaCruz | SC-7392 | IB(1:1000) |
| Recombinant DNA reagent | Anti-V5 (mouse monoclonal) | Bio-Rad | MCA1360 | IB(1:5000) |
| Sequence-based reagent | Anti-Tubulin (mouse monoclonal) | Sigma | T5168 | IB(1:20,000) |
| Peptide, recombinant protein | Anti-Cdc2 (rabbit polyclonal) | SantaCruz | SC-53 | IB(1:5000) |
| Commercial assay or kit | NEBuilder HiFi DNA Assembly Master Mix | NEB | E2621L | |
| Commercial assay or kit | MasterPure Yeast RNA Extraction Kit | Biosearch Technologies | MPY03100 | |
| Commercial assay or kit | NEBNext Poly(A) mRNA Magnetic Isolation Module | NEB | E7490 | |
| Commercial assay or kit | RevertAid Firest Strand cDNA Synthesis Kit | Thermo | K1621 | |
| Commercial assay or kit | Nanopore Direct cDNA Sequencing Kit | Oxford Nanopore | SQK-DCS109 | |
| Commercial assay or kit | MiniION sequencing device | Oxford Nanopore | MIN-101B | |
| Commercial assay or kit | MiniION Flow Cell | Oxford Nanopore | FLO-MIN106D | Chemistry type (R9.4.1) |
| Commercial assay or kit | Microcon Y M-30 Filter Unit | Millipore | 42410 MRCF0R030 | |
| Commercial assay or kit | SPE Column (C18 resin) | Pierce | 89870 | |
| Chemical compound, drug | 3-BrB-PP1 | Abcam | ab143756 | |
| Software, algorithm | Bowtie2 | PMID:22388286 | v2.4.4 | |
| Software, algorithm | Puseq_app | PMID:26492137 | v1 | R script |
| Software, algorithm | MaxQuant | PMID:27809316 | v1.6.12.0 | |

*Continued on next page*

*Continued*

| Reagent type (species) or resource | Designation | Source or reference | Identifiers | Additional information |
|---|---|---|---|---|
| Software, algorithm | Perseus | PMID:27348712 | v1.6.15.0 | |
| Software, algorithm | Minimap2 | *Li, 2018* | | |
| Software, algorithm | GffCompare | *Pertea and Pertea, 2020* | | |

## Strain construction

Strains used in this study are listed in *Table 1* and are available from the authors upon request. Standard yeast genetic techniques and media were used as previously described (*Moreno et al., 1991*). All liquid cultures were grown in YES media at 30 °C unless otherwise stated. Cells were plated onto YEA plus the required selective agents during strain construction. Deletion of region A of *RTS1* was conducted using overlapping primers lacking region A (A30/A31; see *Table 2*) to amplify the pAW8-*RTS1-ura4* plasmid to create pAW8-*RTS1_A∆-ura4*. This plasmid was then transformed into the relevant strain to introduce RTS1_A∆ via the recombination mediated cassette exchange (RMCE) method *Watson et al., 2008* followed by genetic crosses to create the final strains BAY237-BAY242. To create the *RTS1* slippage assay strains, a fragment containing a region of *ura4* containing the 20 bp tandem repeat of *ura4-sd20* (*Iraqui et al., 2012*) was synthesised by Eurofins and digested with *Stu*I and *Dra*III for insertion into pAW8-RTS1-ura4 or pAW8-RTS1_A∆-ura4 plasmids to create the plasmids pAW8-RTS1-ura4-sd20 or pAW8-RTS1_A∆-ura4-sd20. These plasmids were then transformed into the relevant strain to introduce the constructs via the recombination mediated cassette exchange (RMCE) method (*Watson et al., 2008*) to create the strains BAY144 and BAY249, respectively.

The BirA[TbID] C-terminally tagged Rtf2 was constructed via construction of plasmid pAW8-3HA-TurboID:KAN. A 3HA-BirA[TbID] encoding sequence was amplified using P11/P12 primers and the PCR fragment and pAW8 plasmid digested with *Sph*I/*Asc*I and ligated. The resulting plasmid, pAW8-3HA-TurboID:KAN was transformed into an *rtf2* C-terminal tagging base strain to create BAY246 via the RMCE method (*Watson et al., 2008*). The same method is used to C-terminally tag Rtf2 with 3 HA alone, via transformation of pAW8-3HA:KAN to create BAY182.

To create *rtf1∆int* strains, NEBuilder HiFi DNA Assembly Master Mix (Cat #E2621L) was used to combine *rtf1* 5'UTR (P1/P2) with *rtf1* CDS (synthesised by Eurofins, P3/P4) and a hygromycin marker cassette (P5/P6) was amplified with the indicated primers and cloned into *Spe*I/*Sph*I digested pAW8 to create pAW8-rtf1-CDS:hphMX6. The construct was then amplified and transformed into cells via linear fragment transformation.

For *rtf1intron2NE* strains, DNA fragments containing 400 bp of *rtf1* 5'UTR and the *rtf1* coding sequence with all introns removed except for intron 2 were synthesised (Integrated DNA technologies). Intron 2 3' (AG) splice site was mutated (to AA). An identical DNA fragment was synthesised but the *rtf1* coding sequence included an N-terminal V5 tag. The V5 tag and *rtf1* sequence were separated by a small amino acid linker (Gly-Ala-Gly-Ala-Gly-Ala). The fragments were cloned into pAW8-hphMX6 restricted with *Sph*I and *Bgl*II to create pAW8-rtf1intron2NE-hphMX6 and pAW8-V5-rtf1intron2NE-hphMX6, respectively. The constructs were removed by restriction with *Sph*I and *Pme*I, gel isolated and transformed into cells via linear fragment transformation.

N-terminal tagging of Rtf1 with the V5 tag was conducted by first introducing the *ura4* gene between *rtf1* 5'UTR and exon 1 and transformation to create a strain containing *ura4:rtf1*. A synthesised DNA sequence containing 400 bp of the 5'UTR followed by the V5 tag, then a small amino acid linker (Gly-Ala-Gly-Ala-Gly-Ala) and finally 400 bp of exon 1 of *rtf1* was digested with *Sal*I and ligated into *Sal*I digested pAW1 plasmid to create pAW1-V5-rtf1. This fragment was then amplified and the linear fragment transformed into the *ura4:rtf1* containing strain, replacing *ura4* with the V5 tag to create BAY285.

## Replication fork slippage assay

*S. pombe* strains containing the *RTS1-ura4-sd20* construct were grown in 10 ml YE containing 1 mg/ml 5-FOA overnight at 30 °C. Cells were washed in 1 ml 5-FOA-free YE and resuspended into 10 ml fresh YE at a density of 2x10$^6$ cells/ml. Cells were grown for two cell cycles before pelleting and

**Table 1.** Strain list.

| Strain | Genotype | Reference |
|---|---|---|
| BAY119 | II::RTS1-ura4-10xrRFB, RTS1Δ::Phleo, rnh201Δ::KAN, rtf2Δ::NAT, cdc6L591G, ade6-704, leu1-32, ura4-d18 | This study |
| BAY120 | II::RTS1-ura4-10xrRFB, RTS1Δ::Phleo, rnh201Δ::KAN, rtf2Δ::NAT, rtf1Δ::HYG, cdc6L591G, ade6-704, leu1-32, ura4-d18 | This study |
| BAY121 | II::RTS1-ura4-10xrRFB, RTS1Δ::Phleo, rnh201Δ::KAN, rtf2Δ::NAT, cdc20_M630F, ade6-704, leu1-32, ura4-d18 | This study |
| BAY122 | II::RTS1-ura4-10xrRFB, RTS1Δ::Phleo, rnh201Δ::KAN, rtf2Δ::NAT, rtf1Δ::HYG, cdc20_M630F, ade6-704, leu1-32, ura4-d18 | This study |
| BAY123 | II::RTS1-ura4-10xrRFB, RTS1Δ::Phleo, rnh201Δ::KAN, rtf1Δ::HYG, cdc20M630F, ade6-704, leu1-32, ura4-d18 | *Naiman et al., 2021* |
| BAY124 | II::RTS1-ura4-10xrRFB, RTS1Δ::Phleo, rnh201Δ::KAN, cdc20M630F, ade6-704, leu1-32, ura4-d18 | *Naiman et al., 2021* |
| BAY125 | II: RTS1-ura4-10xrRFB, RTS1Δ::Phleo, rnh201Δ::KAN, rtf1Δ::HYG, cdc6L591G, ade6-704, leu1-32, ura4-d18 | *Naiman et al., 2021* |
| BAY126 | II::RTS1-ura4-10xrRFB, RTS1Δ::Phleo, rnh201Δ::KAN, cdc6L591G, ade6-704, leu1-32, ura4-d18 | *Naiman et al., 2021* |
| BAY144 | II::Rura4sd20-10xrRFB, RTS1Δ::Phleo, rtf1Δ::NAT, ade6-704, leu1-32, ura4-d18 | This study |
| BAY146 | II::Rura4sd20-10xrRFB, RTS1Δ::Phleo, ade6-704, leu1-32, ura4-d18 | This study |
| BAY176 | II::Rura4sd20-10xrRFB, RTS1Δ::Phleo, rtf2Δ::NAT, ade6-704, leu1-32, ura4-d18 | This study |
| BAY182 | rtf2-3HA:KAN, ade6-704, leu1-32, ura4-d18 | This study |
| BAY237 | II::RTS1_AΔ-ura4-10xrRFB, RTS1Δ::Phleo, rnh201Δ::KAN, cdc6L591G, ade6-704, leu1-32, ura4-d18 | This study |
| BAY238 | II::RTS1_AΔ-ura4-10xrRFB, RTS1Δ::Phleo, rnh201Δ::KAN, cdc20M630F, ade6-704, leu1-32, ura4-d18 | This study |
| BAY239 | II::RTS1_AΔ-ura4-10xrRFB, RTS1Δ::Phleo, rnh201Δ::KAN, rtf1Δ::HYG, cdc6L591G, ade6-704, leu1-32, ura4-d18 | This study |
| BAY240 | II::RTS1_AΔ-ura4-10xrRFB, RTS1Δ::Phleo, rnh201Δ::KAN, rtf1Δ::HYG, cdc20M630F, ade6-704, leu1-32, ura4-d18 | This study |
| BAY241 | II::RTS1_AΔ-ura4-10xrRFB, RTS1Δ::Phleo, rnh201Δ::KAN, rtf2Δ::NAT, cdc20_M630F, ade6-704, leu1-32, ura4-d18 | This study |
| BAY242 | II::RTS1_AΔ-ura4-10xrRFB, RTS1Δ::Phleo, rnh201Δ::KAN, rtf2Δ::NAT, cdc6L591G, ade6-704, leu1-32, ura4-d18 | This study |
| BAY246 | rtf2-3HA:TurboID:KAN, ade6-704, leu1-32, ura4-d18 | This study |
| BAY249 | II::RTS1_AΔura4sd20-10xrRFB, RTS1Δ::Phleo, rtf1Δ::HYG, ade6-704, leu1-32, ura4-d18 | This study |
| BAY251 | II::RTS1_AΔura4sd20-10xrRFB, RTS1Δ::Phleo, ade6-704, leu1-32, ura4-d18 | This study |
| BAY252 | II::RTS1_AΔura4sd20-10xrRFB, RTS1Δ::Phleo, rtf2Δ::NAT, ade6-704, leu1-32, ura4-d18 | This study |
| BAY253 | II::RTS1_AΔura4sd20-10xrRFB, RTS1Δ::Phleo, rtf2Δ::NAT, rtf1Δ::HYG, ade6-704, leu1-32, ura4-d18 | This study |
| BAY267 | rtf2-3HA-TurboID:KAN, cdc2asM17, II::RTS1-ura4-10xrRFB, ade6-704, leu1-32, ura4-d18 | This study |
| BAY268 | mud1Δ::HYG, ade6-704, leu1-32, ura4-d18 | This study |
| BAY269 | mud1Δ::HYG, rtf2-3HA:KAN, ade6-704, leu1-32, ura4-d18 | This study |
| BAY273 | mud1Δ::HYG, rtf2Δ::NAT, ade6-704, leu1-32, ura4-d18 | This study |
| BAY278 | II::RTS1-ura4sd20-10xrRFB, rtf1Δint:HYG, RTSΔ::Phleo, ade6-704, leu1-32, ura4-d18 | This study |
| BAY279 | II::RTS1-ura4sd20-10xrRFB, rtf1Δint:HYG, RTSΔ::Phleo, rtf2Δ::NAT, ade6-704, leu1-32, ura4-d18 | This study |
| BAY280 | II::RTS1-ura4-10xrRFB, RTS1Δ::Phleo, rtf1Δint:HYG, rnh201Δ::KAN, cdc20M630F, ade6-704, leu1-32, ura4-d18 | This study |
| BAY281 | II::RTS1-ura4-10xrRFB, RTS1Δ::Phleo, rtf1Δint:HYG, rtf2Δ::NAT, rnh201Δ::KAN, cdc20M630F, ade6-704, leu1-32, ura4-d18 | This study |
| BAY282 | II::RTS1-ura4-10xrRFB, RTS1Δ::Phleo, rtf1Δint:HYG, rnh201Δ::KAN, cdc6L591G, ade6-704, leu1-32, ura4-d18 | This study |
| BAY283 | II::RTS1-ura4-10xrRFB, RTS1Δ::Phleo, rtf1Δint:HYG, rtf2Δ::NAT, rnh201Δ::KAN, cdc6L591G, ade6-704, leu1-32, ura4-d18 | This study |
| BAY285 | V5-rtf1, ade6-704, leu1-32, ura4-d18 | This study |
| BAY286 | V5-rtf1, rtf2Δ::NAT, ade6-704, leu1-32, ura4-d18 | This study |
| AW2280 | rtf1 intron2NE-hphMX6, ade6-704, leu1-32, ura4D18 | This study |

*Table 1 continued on next page*

*Table 1 continued*

| Strain | Genotype | Reference |
|---|---|---|
| AW2284 | *V5-rtf1 intron2NE-hphMX6, ade6-704, leu1-32, ura4D18* | This study |
| AW2309 | *V5-rtf1 intron2NE-hphMX6, rtf2::natMX6, ade6-704, leu1-32, ura4D18* | This study |
| AW2324 | *II:Rura4sd20-10XrRFB, RTS1::pleoMX6, rtf1 intron2NE:hphMX6, ade6-704, leu1-32, ura4D18* | This study |
| AW2327 | *II:Rura4sd20-10XrRFB, RTS1::pleoMX6, rtf1 intron2NE:hphMX6, rtf2::natMX6, ade6-704, leu1-32, ura4D18* | This study |

re-suspending in 1 ml ddH$_2$O. A total of 100 µl of cells were then plated in appropriate dilutions onto 2 YEA plates and 2 YNBA plates containing appropriate amino acids minus uracil. Plates were then incubated at 30 °C for 3–5 days. The numbers of colonies were counted and the average mean was taken between each of the two plates. Reversion frequency of Ura$^+$ colonies was then calculated (*Supplementary file 7*), taking into consideration the dilutions plated between YEA plates and YNBA plates lacking uracil.

## Protein extraction and western blotting

*S. pombe* strains grown to logarithmic phase were collected and 5x10$^7$ cells were washed and re-suspended in 200 µl ml 20% trichloroacetic acid (TCA). Samples treated with cycloheximide to block protein synthesis were treated at a final concentration of 100 µg/ml before collection at the indicated time points after addition of drug. Cells were then lysed using glass beads and a Ribolyser (Fast Prep Hybaid, Cat #FP120) for 3x30 s at 6.5 m/s. Glass beads were removed and the pellet re-suspended in 200 µl 1 X Protein Loading Buffer (250 mM Tris pH 6.8, 8% SDS, 20% glycerol, 20% β-mercaptoethanol, and 0.4% bromophenol blue) and boiled at 95 °C for 10 min.

Proteins were separated using SDS-PAGE followed by transfer onto a nitrocellulose membrane (Amersham, Cat #45004003) using the Invitrogen XCell II Blot Module. Membranes were then blocked in 5% Milk (dissolved in PBST (PBS + 0.1% (v/v) Tween)) for 1 hr before incubation with the relevant primary antibody diluted in 5% Milk PBST overnight at 4 °C. (anti-HA SantaCruz SC-7392, 1:1000; anti-V5 Bio-Rad MCA1360, 1:5000; anti-Tubulin Sigma T5168, 1:20,000; anti-Cdc2 SantaCruz SC-53, 1:5000). Membranes were then washed 3 x with PBST for 5 min each at room temperature before incubation with a horse radish peroxidase-conjugated secondary antibody (HA/V5/Tubulin: Rabbit anti-mouse Dako P0161. Cdc2: swine anti-rabbit Dako P0217) at room temperature for 1 hr followed

**Table 2.** Primer list.

| Name | Sequence |
|---|---|
| A30 | GGAGGTTGAGTGTGGGACGTTTCTGCCATACCCTTTTTAAGT |
| A31 | GGTATGGCAGAAACGTCCCACACTCAACCTCCCAAT |
| P1 | ATTATACGAAGTTATGCATGGTTTGATATGAGGCAGATAC |
| P2 | TTCCTTGCATAATAATGTTCACTTGTCTGAAG |
| P3 | GAACATTATTATGCAAGGAAAAAACAATTTAAG |
| P4 | CGCTGGCCGGCTAGCATAAATCATCGGC |
| P5 | TTTATGCTAGCCGGCCAGCGACATGGAG |
| P6 | ATACCATATACGAAGTTATACGACAGCAGTATAGCGACCAG |
| P7 | GGAGCAAACGACATTATCAC |
| P8 | CATCACGATGGTTATCAGAC |
| P9 | CTATGGACAGCAGATGCTTG |
| P10 | GCGGTGTAAGAATCATGTAA |
| P11 | GAAGTTATGCATGCTCTACCCGTATGATGTTCCGGA |
| P12 | AGCTGCGGCGCGCCTCACTTTTCGGCAGACCGCAGAC |

by a further 3 x PBST washes. Proteins were detected using the Western Lightning Plus-ECL chemi-luminescent substrate (Perkin Elmer, Cat #NEL103001EA) and autoradiograph film processed in an X-ray film developer.

## Polymerase-usage sequencing

The published protocol was used for library generation and the published pipeline (that uses Bowtie2 to align sequence files and convert mapped reads to count files) was followed (*Keszthelyi et al., 2015*). Analysis of the read counts was conducted using the published R script to obtain polymerase usage and polymerase bias information (*Keszthelyi et al., 2015*).

## Nanopore direct cDNA-sequencing

Total RNA was extracted from 10 ml of logarithmically grown *S. pombe* cells using the Master-Pure Yeast RNA Extraction Kit (Cat #MPY03100). PolyA-mRNA was isolated using NEBNext Poly(A) mRNA Magnetic Isolation Module (Cat #E7490). Purified PolyA-mRNA was then used to prepare cDNA sequencing libraries using the Nanopore Direct cDNA Sequencing Kit according to manufacturer's instructions (Cat #SQK-DCS109). The cDNA libraries were sequenced using the MinION sequencing device (Cat #MIN-101B) and associated flow cells (Cat #FLO-MIN106D). Sequenced reads were processed using the Nanopore Technology Reference Isoforms pipeline (*Wright et al., 2022*). This pipeline sorts and aligns reads to the reference genome producing alignment BAM files and consensus transcript annotation GFF files. Calculations of intron retention (IR) were based on an alignment threshold of>70% for reads to each intronic and flanking exonic sequence. An intron was only counted as an IR event if the subsequent exonic sequence was also present. It should be noted that this analysis does not necessarily identify all mis-splicing events.

Testing intron retention levels via PCR amplification was conducted using RevertAid First Strand cDNA Synthesis Kit (Thermo, Cat #K1621) to create cDNA from mRNA. Amplification of *rtf1* intron 2 was carried out using primer pair P7/P8, and intron 6 amplified using P9/P10. PCR product were purified using a QIAquick PCR Purification Kit (Qiagen, Cat #28104) before running on a 2% agarose gel.

## Cell synchronisation

*S. pombe* strains containing the *cdc2asM17* ATP-analogue sensitive allele (*Singh et al., 2021*) were grown O/N at 28 °C. Once cell density reached $2.5 \times 10^6$ cells/ml, 1:1000 volume of 3-BrB-PP1 (2 mM) was added to the culture and incubated at 28 °C for a further 3 hr to synchronise in G2. Cultures were then filtered using a vacuum flask filter unit. Cells collected on the filter paper (0.22 µm, Millipore, Cat #N8645) were then washed three times by addition of fresh YE media and filtration. The cell coated filter paper was then placed into pre-warmed fresh YE media before resuspension by shaking. Cells were grown at 28 °C for sample collection in S phase.

## Affinity capture of biotinylated proteins

Cells were grown in YE media and 50 µg biotin was added 3 hr before cells were harvested. Capture of biotinylated proteins from *rtf2*⁺ (untagged) and *rtf2-3HA-BirA* samples was conducted as previously described (*Larochelle et al., 2019*). After incubation with streptavidin Sepharose beads and subsequent washing steps, 50 µl SDS protein loading buffer (with added *d*-Desthiobiotin to a final concentration of 2.5 mM) was added to the beads and boiled at 95 °C for 10 min before cooling at RT for 10 min.

## Filter-aided sample preparation (FASP) for MS samples

Beads were separated from supernatant as before and 50 µl of the eluate was added to 333 µl of FASP Urea Solution (8 M Urea, 0.1 M Tris/HCl, pH 8.5, 30 mM DTT) and transferred to a Microcon Y M-30 (Millipore, 42410) filter unit. Filter units were centrifuged at 14,000 x *g* for 15 min followed by addition of another 200 µl FASP Urea solution and centrifuged again. Flow through was discarded before addition of 100 µl FASP IAA (8 M Urea, 50 mM iodoacetamide (IAA)) solution to each filter unit and incubation at RT for 20 min followed by centrifugation as before. Filter units then received 100 µl FASP Urea solution before centrifuging as before (this step is repeated twice for a total of three times). Flow through was discarded and 100 µl ABC buffer (50 mM Ammonium bicarbonate) was added to each filter unit before centrifuging at 14,000 x *g* for 10 min (this step is repeated twice for a total of

three times). Flow through was discarded and 40 µl digestion solution was added to each filter unit and incubated overnight in a wet chamber. Filter units were transferred to fresh collection tubes and centrifuged at 14,000 x *g* for 10 min. Following this 40 µl ABC buffer was added and centrifuged again in the same conditions.

## Desalting of peptides with solid phase extraction (SPE) columns

Filtrate from the FASP digestion was acidified by addition of 100 µl Buffer A* (5% Acetonitrile, 3% Trifluoroacetic Acid). SPE columns containing 8 mg C-18 resin (Pierce, 89870) were activated by addition of 100 µl 50% methanol and centrifuged at 1500 x *g* for 1 min, and flow through discarded (repeated once for a total of two times). The columns were then equilibrated twice by addition of 200 µl Buffer A (5% Acetonitrile, 0.1% Formic Acid) and centrifuged as before discarding the flow through after each spin. Each sample was then added to the equilibrated spin column and centrifuged as before, washed two times with 200 µl Buffer A before elution of the desalted peptides with 100 µl of Buffer B (80% Acetonitrile, 0.1% Formic Acid). Eluates were concentrated in a speedvac to a volume of 1–2 µl and resuspended in 20 µl of Buffer A before LC-MS/MS analysis.

## LC-MS/MS run and analysis

Desalted peptide samples were analysed by a reversed-phase capillary nano liquid chromatography system (Ultimate 3000, Thermo Fisher Scientific) connected to a Q Exactive HF mass spectrometer (Thermo Scientific). Samples were injected and concentrated on a trap column (PepMap100 C18, 3 µm, 100 Å, 75 µm i.d. x 2 cm, Thermo Scientific) equilibrated with 0.05% trifluoroacetic acid in water. After switching the trap column inline, LC separations were performed on a capillary column (Acclaim PepMap100 C18, 2 µm, 100 Å, 75 µm i.d. x 25 cm, Thermo Scientific) at an eluent flow rate of 300 nl/min. Mobile phase A contained 0.1% formic acid in water, and mobile phase B contained 0.1% formic acid in 80% acetonitrile, 20% water. The column was pre-equilibrated with 5% mobile phase B and peptides were separated using a gradient of 5–44% mobile phase B within 40 min. Mass spectra were acquired in a data-dependent mode utilising a single MS survey scan (m/z 350–1650) with a resolution of 60,000 in the Orbitrap, and MS/MS scans of the 15 most intense precursor ions with a resolution of 15,000. HCD-fragmentation was performed for all ions with charge states of 2+to 5+using anormalized collision energy of 27 and isolation window of 1.4 m/z. The dynamic exclusion time was set to 20 s. Automatic gain control (AGC) was set to $3\times10^6$ for MS scans using a maximum injection time of 20ms. For MS2 scans the AGC target was set to $1\times10^5$ with a maximum injection time of 25ms.

MS and MS/MS raw data were analysed using the MaxQuant software package (version 1.6.12.0) with an implemented Andromeda peptide search engine (*Tyanova et al., 2016*). Data were searched against the FASTA formatted Uniprot reference proteome database of *Schizosaccharomyces pombe* (UniprotKB UP000002485). Perseus software (version 1.6.15.0) was used to determine biologically significant hits as calculated by the Welch's T-test.

## Acknowledgements

We thank Dr. Benno Kuropka from BioSupraMol, Freie Universität Berlin for performing the LC-MS runs. AMC acknowledges Welcome Trust award 110047/Z/15/Z.

## Additional information

### Funding

| Funder | Grant reference number | Author |
| --- | --- | --- |
| Wellcome Trust | 11047/Z/15/Z | Antony M Carr |

The funders had no role in study design, data collection and interpretation, or the decision to submit the work for publication. For the purpose of Open Access, the authors have applied a CC BY public copyright license to any Author Accepted Manuscript version arising from this submission.

## Author contributions
Alice M Budden, Conceptualization, Resources, Data curation, Formal analysis, Validation, Investigation, Visualization, Methodology, Writing - original draft; Murat Eravci, Data curation, Methodology, Writing – review and editing; Adam T Watson, Investigation, Methodology, Writing – review and editing; Eduard Campillo-Funollet, Data curation, Validation, Visualization, Writing – review and editing; Antony W Oliver, Formal analysis, Visualization, Writing – review and editing; Karel Naiman, Conceptualization, Data curation, Supervision, Investigation, Writing – review and editing; Antony M Carr, Conceptualization, Formal analysis, Supervision, Funding acquisition, Project administration, Writing – review and editing

## Author ORCIDs
Antony W Oliver (ID) http://orcid.org/0000-0002-2912-8273
Karel Naiman (ID) https://orcid.org/0000-0003-3594-1516
Antony M Carr (ID) http://orcid.org/0000-0002-2028-2389

## Decision letter and Author response
Decision letter https://doi.org/10.7554/eLife.78554.sa1
Author response https://doi.org/10.7554/eLife.78554.sa2

## Additional files

### Supplementary files
- Supplementary file 1. List of significant Mass spec. hits.
- Supplementary file 2. GO terms -S phase experiment.
- Supplementary file 3. GO terms -MMS experiment.
- Supplementary file 4. Transcript types.
- Supplementary file 5. Intron retention calculations.
- Supplementary file 6. Gene list.
- Supplementary file 7. Data for slippage assays.
- MDAR checklist

### Data availability
Sequence data is available under GEO dataset GSE192344.

The following dataset was generated:

| Author(s) | Year | Dataset title | Dataset URL | Database and Identifier |
|---|---|---|---|---|
| Budden AM, Eravci M, Campillo-Funollet E, Naiman K, Carr AM | 2021 | *Rtf2* is Important for Replication Fork Barrier Activity of *RTS1* via Splicing of *Rtf1* | https://www.ncbi.nlm.nih.gov/geo/query/acc.cgi?acc=GSE192344 | NCBI Gene Expression Omnibus, GSE192344 |

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
