## [Editor Report]

This valuable work reports a new regulation of the RNA splicing of a fission yeast gene necessary for the DNA replication fork barrier. Using molecular genetic approaches combined with replication fork profiling analysis, the authors provided significant evidence that the Rtf2 is a splicing regulator specific to the *rtf1* gene essential for the fork barrier. The study is of interest to researchers in the fields of DNA replication and homologous recombination as well as RNA splicing.

---

## [Decision Letter]

**Decision letter after peer review:**

Thank you for submitting your article "*S. pombe Rtf2* is Important for Replication Fork Barrier Activity of *RTS1* via Splicing of *Rtf1*" for consideration by *eLife*. Your article has been reviewed by 3 peer reviewers, including Akira Shinohara as Reviewing Editor and Reviewer #1, and the evaluation has been overseen by Jessica Tyler as the Senior Editor. The following individuals involved in the review of your submission have agreed to reveal their identity: Sarah Lambert (Reviewer #2); Jie Ren (Reviewer #3).

Essential revisions:

Importantly, to strengthen your conclusion in the paper, we recommend one experiment described below. And also you see the comments from three independent reviewers.

Importantly, you showed expression of Rtf1 protein with an insertion of 15 amino acids from the second intron in the rtf2 mutant (Figure 3E, page 12, lines 487-488), which is non-functional for activity at RTS1. Please demonstrate that Rtf1 with extra 15 amino acids from the intron 2 (e.g. by inserting the 15 amino acid sequence in Rtf1delta-int construct in Figure 4) is NOT functional either by in vivo complementation assay (Pu-seq and/or ura4 reversion assay), or ChIP to the RTS1 site, or by in vitro biochemical analysis of DNA binding for purified wild-type Rtf1 protein and Rtf1 with the extra 15 amino acids.

*Reviewer #1 (Recommendations for the authors):*

Please provide results that indicate the molecular mechanism of Rtf2 controlling the splicing of Rtf1 RNA.

1. Although the authors showed the splicing of only the second (and sixth) intron of Rtf1 RNA is defective in the rtf2 mutant (Figure 3D), the authors introduced an intronless RTF1 gene rather than the RTF1 without the second intron for their complementation assay (Figure 4). It would be important to check whether Rtf1 cDNA with intron2 does not complement the rtf1 deletion and/or rtf2 deletion.

2. In the same line to point #1, the authors need to introduce the second intron of the RTF1 gene to a reporter gene to check the dependency of the intron removal on the Rtf2.

3. The authors showed the expression of Rtf1 protein with 15 amino acids from the second intron in the rtf2 mutant (Figure 3E, page 12, lines 487-488), which is non-functional for RTS1 activity. It is important to confirm this claim by experimentation as in point #1. In another word, it is critical to show that Rtf1 with extra 15 amino acids from the intron 2 is NOT functional either by simple complementation assay or biochemical analysis of DNA binding for purified wild-type Rtf1 protein and Rtf1 with the extra 15 amino acids.

4. An RT-PCR product for the removal of Rtf2 second intron in the rtf2 mutant migrates a bit faster on the gel than the product of the "retained" (Figure 3D). The authors need to confirm the DNA sequence of the retained products in both strains.

---

## [Author Response]

Essential revisions:Importantly, to strengthen your conclusion in the paper, we recommend one experiment described below. And also you see the comments from three independent reviewers.Importantly, you showed expression of Rtf1 protein with an insertion of 15 amino acids from the second intron in the rtf2 mutant (Figure 3E, page 12, lines 487-488), which is non-functional for activity at RTS1. Please demonstrate that Rtf1 with extra 15 amino acids from the intron 2 (e.g. by inserting the 15 amino acid sequence in Rtf1delta-int construct in Figure 4) is NOT functional either by in vivo complementation assay (Pu-seq and/or ura4 reversion assay), or ChIP to the RTS1 site, or by in vitro biochemical analysis of DNA binding for purified wild-type Rtf1 protein and Rtf1 with the extra 15 amino acids.

We have carried out an experiment where we express the Rtf2 with the extra 15 amino acids and demonstrate this is non-functional. It is now presented in a new Supplementary Figure.

Reviewer #1 (Recommendations for the authors):Please provide results that indicate the molecular mechanism of Rtf2 controlling the splicing of Rtf1 RNA.

We believe this to be beyond the scope of the manuscript (and the intellectual expertise of the laboratory).

1. Although the authors showed the splicing of only the second (and sixth) intron of Rtf1 RNA is defective in the rtf2 mutant (Figure 3D), the authors introduced an intronless RTF1 gene rather than the RTF1 without the second intron for their complementation assay (Figure 4). It would be important to check whether Rtf1 cDNA with intron2 does not complement the rtf1 deletion and/or rtf2 deletion.

We did not perform this experiment. The cDNA sequencing clearly shows that the major form of rtf1 mRNA in rtf2D cells contains only intron 2. Our demonstration (new supplementary figure) that the Rtf1 which contains the resulting additional amino acids is non-functional confirms the conclusions we draw from the presented data.

2. In the same line to point #1, the authors need to introduce the second intron of the RTF1 gene to a reporter gene to check the dependency of the intron removal on the Rtf2.

Our data suggest that introns that are retained in rtf2D cells when the intron is adjacent to an exon with a low G/C content. A reasonable hypothesis would be that Rtf2-depenedance is influenced by the downstream exon. Thus, we would predict that simply introducing an intron 2 into a random exon would not necessarily result in retention.

3. The authors showed the expression of Rtf1 protein with 15 amino acids from the second intron in the rtf2 mutant (Figure 3E, page 12, lines 487-488), which is non-functional for RTS1 activity. It is important to confirm this claim by experimentation as in point #1. In another word, it is critical to show that Rtf1 with extra 15 amino acids from the intron 2 is NOT functional either by simple complementation assay or biochemical analysis of DNA binding for purified wild-type Rtf1 protein and Rtf1 with the extra 15 amino acids.

A complementation experiment has been done and is presented in a new supplementary figure.

4. An RT-PCR product for the removal of Rtf2 second intron in the rtf2 mutant migrates a bit faster on the gel than the product of the "retained" (Figure 3D). The authors need to confirm the DNA sequence of the retained products in both strains.

The sequence of the retained product is evident from the sequencing of the cDNA. We do not understand why sequencing the product of a PCR reaction would add any further information.